# The adapted French version of the Academic and Athletic Identity Scale (AAIS-FR): Evidence of validity and reliability and relationships with sport well-being

**Solène Lefebvre du Grosriez[1]ʘ, Sandrine Isoard-Gautheur[1]ʘ, Mariya Yukhymenko-Lescroart[2], Philippe Sarrazin[1]ʘ***

**1** Univ. Grenoble Alpes, SENS, Grenoble, France, **2** California State University, Fresno, CA, United States of America

ʘ These authors contributed equally to this work.
* philippe.sarrazin@univ-grenoble-alpes.fr

## Abstract

### Background

Effectively managing their athletic and academic projects is a major challenge for student-athletes. The salience of the identity they develop in each of these contexts can affect their well-being and is therefore an important variable to consider. Examining these mechanisms in countries and student-athlete support systems other than the United States is also important.

### Aim

This study aims to both translate and evaluate the psychometric properties of a French version of the Academic and Athletic Identity Scale, the AAIS-FR, and to examine the additive and interactive relationships of the two identities with sport burnout and engagement.

### Methods

Participants were 359 French student-athletes (50.42% female) who were competing at various levels (ranging from regional to international).

### Results

Results from analyses using a slightly modified version of the original scale provided evidence of construct (i.e., factor structure) and concurrent (i.e., expected relationships between both identities and several correlates identified in previous work) validity, as well as reliability (i.e., internal consistency) and invariance across gender and sport competition levels of the AAIS-FR. In addition, regression analyses revealed a favourable relationship between athletic identity and sport well-being (i.e., positive with engagement and negative with burnout), no relationship between academic identity and sport well-being, and no interaction between the two identities.

**Data Availability Statement:** All relevant data are within the paper and its Supporting Information files.

**Funding:** The author(s) received no specific funding for this work.

**Competing interests:** The authors have declared that no competing interests exist.

## Conclusion

While further research is needed to provide additional evidence for the validity of the AAIS-FR, researchers can still use this tool to measure the salience of the two identities of French-speaking student-athletes.

## Introduction

Student-athletes (SAs) are expected to excel in both the athletic and academic contexts [1]. Previous research has highlighted that SAs were at increased risk for impaired well-being due to the demands of both contexts [2]. However, other work has not reported such a decrease in SA well-being and has even highlighted the positive effects of combining these dual commitments [1, 3]. The impact of dual projects on well-being appears to be related to how SAs identify with each context (e.g., [4, 5]). Therefore, research that examines SAs' dual identities is needed to promote their well-being. To this end, Yukhymenko-Lescroart [6] developed and validated the Athletic and Academic Identity Scale (AAIS). Due to its promising psychometric properties, widespread use in English-speaking countries, and availability in different languages versions, some authors have recently proposed the AAIS as a potential gold standard in dual career research. They have encouraged future research to examine its cross-cultural psychometric validity [7]. The same authors also called for studies to be conducted in contexts other than the North American university campus to explore the impact of structural differences on the development of SAs' identity and well-being. The purpose of this study is precisely to propose a French version of the AAIS and to examine its construct and concurrent validity, as well as its relationship with sport well-being (i.e., athlete engagement and burnout) among French SAs.

### Student-athlete identity

Identity is a fundamental psychological process that results in a subjective interpretation of who we are based on our socio-demographic features, personal characteristics, commitment, and group membership [8, 9]. According to Erikson [10], an individual's identity is multidimensional in the sense that it is made up of a mosaic of specific identities built around the key social roles an individual plays (e.g., athlete, student, worker, child). Each of these identities may be more or less salient to individuals, depending on the importance they attach to some over others. This identity salience is related to individuals' levels of commitment to their various roles [11], which may fluctuate over time [12, 13]. In addition, identity salience provides a cognitive framework that guides and organizes information processing and determines how individuals should feel or behave in a given situation [11].

Many elite athletes are involved in both competitive sport and higher education. Consistent with work on identity development, they should develop both academic and athletic identities that align with their two primary roles [5, 7]. Conceptually, "academic identity can be defined as the centrality of being a student to the sense of self. Likewise, athletic identity can refer to the centrality of being an athlete to the sense of self" [14] (p.3). While SAs may develop both identities, it is also possible that one may be more salient for some [6]. For example, the athletic identity may be highly salient for some SAs and block the development of their academic identity, a phenomenon known as identity foreclosure [15].

Ultimately, understanding how dual projects affect SAs' well-being requires assessing how they identify with each of the two contexts (i.e., athletic and academic). In other words, it is

necessary to carefully examine multidimensional identities, not just one or the other [5, 7]. The studies that have done so are relatively few in number [7] and have used different instruments with more or less satisfactory psychometric qualities (see for a review [6, 7]), making it difficult to compare and generalize the results across studies. The AAIS [6] is an 11-item scale specifically designed to measure both academic and athletic identities in a brief and specific format. Respondents are asked to what extent each of the presented characteristics (e.g., being athletic) is central to their sense of self. They use a response scale consisting of several concentric circles representing different degrees of centrality of self-identification, from the value 6, the central circle anchored by "very central core to who I really am" to the value 1 falling outside the larger circle anchored by "not central to who I really am". In two studies conducted with a sample of National Collegiate Athletic Association (NCAA) SAs, Yukhymenko-Lescroart [6] provided evidence for the content and factorial validity of the AAIS, as well as its reliability and invariance across gender and sport competition level [6]. Since its development, the AAIS has been widely used in English-speaking countries such as the United States [14, 16, 17], United Kingdom [18] and Australia [5]. More recently, the instrument has been translated and validated with Japanese [19, 20] and Brazilian [21] participants. Given this extensive use and promising psychometric properties, a recent systematic literature review recommended examining the cross-cultural psychometric validity of the AAIS [7]. To validate the AAIS in another language, it is necessary to know the correlates of academic and athletic identities.

## Correlates of academic and athletic identities

Numerous correlates of athletic or academic identity have been documented (i.e., variables that may have reciprocal causality with identity), with associations whose effect sizes are generally small to moderate [22]. For example, several studies have confirmed Brewer et al.'s [8] hypothesis that athletes competing at elite and advanced levels should endorse a higher athletic identity than those competing at lower levels [5, 6, 20]; for reviews see [7, 22]. Reaching international or Olympic levels requires a greater commitment of time, energy and resources than reaching lower levels. Athletes at the international or Olympic level also tend to receive more social support and recognition than those at lower levels. Greater commitment and social support probably explain why athletes who reach the highest levels of performance have a stronger identification with athletic role than those at lower levels. In turn, this greater commitment and better athletic outcomes, as well as a higher value placed on the athletic experience, generally lead athletes with a strong athletic identity to experience greater satisfaction with their athletic outcomes than those with a lower athletic identity [23]. Furthermore, based on the assumption that sport conforms more to male than to female gender norms and roles [24], several studies have reported that male athletes adopt a more athletic identity than female athletes [25, 26]. However, the findings have not always been consistent [6], and recent work has reported no differences in athletic identity between males and females [13, 27], perhaps reflecting the evolution of gender stereotypes and societal expectations regarding women's place in sport [28].

Previous literature has also shown that academic identity is related to various variables such as gender, academic investment, achievement, and satisfaction (see for a review [29]). For example, studies report that girls have a more salient academic identity than boys [13, 26, 27]. However, this difference does not appear in all studies [5, 6]. In addition, students who identify strongly with their role as a student put more effort into their studies [16], have higher academic commitment [30], and get higher grades [13, 17, 19, 27, 31] than those who identify weakly.

## Student-athlete identity and sport well-being

Well-being is more than the absence of mental illness [32]. It is a state that allows for optimal functioning in the global life or in a specific context, which makes it possible to study it at a global and contextual level [33]. Research that has examined the relationship between identity and well-being among SAs has been somewhat inconsistent, in part because of the variety of well-being constructs that have been used and the levels of analysis (global or contextual) at which they have been examined, making it difficult to compare results across studies. A growing body of research suggests that the well-being of athletes is linked to the meaningful contexts in which they develop [33]. For many SAs, elite sport occupies a large part of their lives. This context contains many demands and challenges that can make it a stress factor or a source of satisfaction and fulfilment. Therefore, it seems important to examine SAs' well-being specific to the sport context [4] using indicators that capture both adaptive and maladaptive facets of the experience, such as athlete burnout and athlete engagement [34, 35].

Athlete burnout is generally described as a negative psychological state characterized by a combination of emotional and physical exhaustion, devaluation of sport, and reduced sense of accomplishment [36]. With a few exceptions [37], most studies have reported a negative correlation between the salience of SAs' athletic identity and burnout [38–40], suggesting that endorsing such an identity could help prevent burnout. Because athletic identity is associated with high motivation and energy for sports [8], athletes who are willing to train hard may be more likely to endure psychological and physical discomfort [40]. Additionally, athletes with a strong athletic identity are more likely to possess high levels of mental toughness [40], a psychological characteristic that may protect them from experiencing the symptoms of athlete burnout [41].

While burnout represents a maladaptive cognitive-affective experience, its absence does not necessarily indicate the presence of an adaptive experience. For this reason, some authors recommend assessing both the positive and negative facets of the sport experience [34]. Athlete engagement is generally defined as a positive and relatively enduring cognitive-affective experience in sport, characterized by confidence, dedication, vigor, and enthusiasm [35]. We found only one study that examined the relationship between athletic identity and this construct [42]. This study was conducted among elite Croatian sprinters and found positive correlations between certain dimensions of engagement (dedication, enthusiasm) and athletic identity. Other positive relationships were highlighted in the meta-analysis by Lochbaum et al. [22] between athletic identity and constructs related to certain dimensions of engagement, such as intrinsic motivation and commitment (constructs close to dedication), positive affect, emotions and feelings (close to enthusiasm), self-esteem, worth, and competence (close to confidence). Among SAs, four studies have shown that athletic identity is positively related to constructs closely associated with sport engagement, such as enthusiastic commitment [19], achievement [43] or autonomous [44] motivation, or intrinsic interest in and value of sport [16]. In summary, previous research has shown positive associations between athletic identity, on the one hand, and a higher desire to invest time and effort in sport, self-confidence to achieve goals and succeed in sport, and intrinsic motivation for sport, on the other hand; in short, characteristics of higher engagement.

Not only have there been few studies examining the relationship between SA identities and athletic well-being, but those that exist have used a diversity of approaches, making it complicated to compare results. At least four different approaches have been employed: *isolated* (between athletic identity *or* academic identity and well-being without regard to the other identity; e.g., [38]), *globalized* (between the overall "student-athlete" identity, rather than each of the two identities measured separately; e.g., [45]), *additive* (between both identities and well-being, where each identity may be positively or negatively related to well-being; e.g., [40]), and *interactive* (where one identity may moderate the relationship between the other identity and well-being; e.g., [4])

approaches. The majority of studies have used the isolated approach, focusing only on athletic identity. To the extent that SAs are involved in two primary roles, it is critical to examine how identification with these two roles relates to sport well-being. It is possible that a strong identification with the student role allows for the development of resources and skills in this area that help to enhance sport well-being. It is also possible that a strong identification with the student role creates role and identity conflicts that may hinder sport well-being [27]. To gain a deeper understanding of how the two identities coexist and contribute together to the well-being of SAs, either additive or interactive approaches are needed. To our knowledge, only one study has examined the relationship between the two identities and athlete burnout using the additive method [40]. While athletic identity was negatively related to athlete burnout, as in the majority of previous studies, academic identity was positively related to this variable. In other words, the stronger the SAs' academic identity, the more sport burnout they reported, thus limiting the positive effects of athletic identity on this variable. Finally, we found only one study that used the interactive method to predict sport well-being [4], as indexed by the Sport Mental Health Continuum-Short Form [46]. While athletic identity predicted (negatively) sport well-being, neither academic identity nor the interaction between the two identities was related to this variable. This result suggests that academic identity did not moderate the relationship between athletic identity and sport well-being. However, as this study took place during the second wave of the COVID-19 pandemic, further research is needed to verify that this result is not specific to this study.

## The present study

Based on the reviewed literature, the present study had two main objectives. The first goal was to develop a French version of the AAIS (AAIS-FR), to test its construct validity (i.e., factor structure), reliability (i.e., internal consistency), invariance with gender and sport competition level, as well as its concurrent validity by examining its relationships with gender, level of competition, time spent in athletic and academic contexts, satisfaction with athletic and academic results. The second goal was to examine the relationships between the two identities and SAs sport well-being. As suggested by several authors [34, 35], we measured burnout and engagement which capture two complementary facets of sport well-being, a maladaptive and an adaptive cognitive and affective experience, respectively.

Based on previous results, we expected that (a) the factor structure and reliability of the AAIS-FR would be similar to the original scale, (b) the AAIS-FR would be invariant across gender and sport competition level as the original scale, (c) athletic identity would be positively related to time spent in sport, satisfaction with sport results, and higher levels of sport competition, but not to gender, (d) academic identity would be positively related to time spent on academics, satisfaction with academic results, and would be higher for women than for men, (e) athletic identity would negatively predict athlete burnout and positively predict athlete engagement. No hypotheses were formulated regarding relationships between time dedicated or satisfaction with results in one context (e.g., academic) and identity in the other context (e.g., athletic) due to a lack of empirical data. For the same reasons, no hypotheses were formulated regarding the relationship between academic identity and athlete burnout or engagement, or the interaction between the two identities on sport well-being. However, the additive or interactive effects of the two identities on sport well-being will be examined.

## Methods

### Participants and procedure

As part of a larger project on the well-being of SAs, a convenience sample of 359 French SAs studying in various academic years, across diverse academic disciplines, and competing at

**Table 1. Basic characteristics of the sample.**

| Variables | | Number | Percentage |
|---|---|---|---|
| Gender | Female | 181 | 50.4 |
| | Male | 178 | 49.6 |
| Age | <18 | 26 | 7.2 |
| | 18–19 | 213 | 59.3 |
| | 20–21 | 80 | 22.3 |
| | 22–23 | 30 | 8.4 |
| | >23 | 10 | 2.8 |
| Type of sport | Individual sport (e.g., track and field, gymnastics) | 216 | 60.2 |
| | Dual sport (e.g., judo, tennis) | 42 | 11.7 |
| | Team sport (e.g., ice-hockey, basketball) | 101 | 28.1 |
| Level of competition | Regional | 98 | 27.3 |
| | National | 162 | 45.1 |
| | International | 99 | 27.6 |
| Level of study | First year (undergraduate) | 184 | 51.3 |
| | Second year (undergraduate) | 109 | 30.4 |
| | Third year (undergraduate) | 35 | 9.7 |
| | Fourth year (Master's level) | 22 | 6.1 |
| | Fifth year (Master's level) | 9 | 2.5 |
| Academic discipline | Sport and exercise science | 206 | 57.4 |
| | Other (e.g., engineering, life or earth sciences | 153 | 42.6 |

different levels from a variety of sports was included in the study. These SAs spent an average of 11.70 hours (SD = 6.38) and 25.80 hours (SD = 11.80) per week to athletic and academic participation, respectively. General information about the study sample is provided in Table 1.

All participants were contacted to complete the online survey via an email sent by their university sports service. Participation was voluntary and informed consent was obtained from each participant. The study was completely anonymous, with no identifying information (e.g., address, date of birth). A written consent form was displayed on the first page of the online questionnaire, providing participants with the option to download a hard copy for their records. To access the questionnaire, participants had to tick the box "I agree to participate in this study in a free and informed manner." In the case of under-age students, the form required that their parents (or guardians) be consulted to obtain their consent before they could participate in the study. The study was approved by the first author's university privacy officer before data collection began. Data were collected between October 06, 2020 and November 02, 2020.

### Measure

**Participants and contextual characteristics.** Questions about gender, academic (i.e., major, academic year, number of hours per week spent on academic activities, satisfaction with academic results), and athletic (i.e., type of sport, level of sport competition, number of hours per week spent on athletic activities, satisfaction with athletic results) characteristics of SAs were used in the present study. Time spent on each role was assessed by one item for each context (i.e., "*How many hours per week do you spend on your studies [sport]?*"). Satisfaction with the results for each role was assessed by one item for each context (i.e., "*How satisfied would you say you are with the results of your studies [sport]?*"), with responses provided on a 7-point response format ranging from (1) *not satisfied at all* to (7) *very satisfied.*

**Academic and athletic identities.** A French translation of the 11-item AAIS was used. It consists of two subscales: academic identity (five items, e.g., "*Doing well in school.*") and athletic identity (six items, e.g., "*Being a good athlete.*"). Responses are given on a 6-point scale ranging from (1) *not central to my sense of self* to (6) *very central to my sense of self*. The items are preceded by the sentence "Please indicate how central each of the following characteristics is to your sense of who you really are". The French version of the questionnaire with the introductory part is provided in Supplementary Material. We followed the guidelines for the process of cross-cultural adaptation of self-report measures [47]. Specifically, the original version of the AAIS was independently translated from English into French by three French sport psychology researchers fluent in English. They compared and discussed the translations and agreed on a preliminary French version that retained the basic features of the original questionnaire. Then, during a back-translation stage, the French version was translated into English by a professional English translator who had not been involved in the first translation phase. The two English versions were then compared for inconsistencies. Discussions were held between the researchers and the translator to reach a consensus where there were differences between the two versions.

**Sport engagement.** A French version of the 16-item Athlete Engagement Questionnaire (AEQ) was used to assess the positive side of well-being [35]. The scale consists of four subscales: confidence (four items, e.g., "*I feel capable of success in my sport*"), dedication (four items, e.g., "*I am devoted to my sport*"), vigour (four items, e.g., "*I feel energized when I participate in my sport*"), and enthusiasm (four items, e.g., "*I feel excited about my sport*"). Responses are given on a five-point scale, from (1) *almost never* to (5) *almost always*. Adequate factor structure of the scale: $\chi^2$ (363) = 359.916, df = 98, p < 0.001, CFI = 0.93, SRMR = 0.065, and reliability (ω ranging from 0.85 to 0.90 for the subscales and equal to 0.97 for the total scale) were demonstrated in this study.

**Sport burnout.** The Athlete Burnout Scale (ABO-S), a French 14 items scale measuring athlete burnout [48], was used to assess the negative side of well-being. It consists of three subscales: reduced sense of accomplishment (four items, e.g., "*I am not performing up to my abilities*"), physical exhaustion (five items, e.g., "*I feel physically exhausted*"), and negative feelings (five items, e.g., "*I have negative feelings toward my sport*"). Reponses are given on a five-point scale, from (1) *almost never* to (5) *almost always*. Adequate factor structure of the scale: $\chi^2$ (74, $N = 359$) = 282.916, $p < 0.001$, CFI = 0.90, SRMR = 0.069, and reliability (ω ranging from 0.77 to 0.90 for the subscales and equal to 0.93 for the total scale) were demonstrated in this study.

## Data analysis

All analyses were performed using the R software [49] with the following packages: "psych" [50], "lavaan" [51], "corx" [52], "interactions" [53]. First, the factorial structure was examined using Confirmatory Factor Analysis (CFA) to test whether the 11 items of the AAIS-FR grouped well on two latent variables, without the possibility of cross-loading. The fit of the tested model was assessed by several goodness-of-fit indices [54], including chi-square ($\chi^2$), Comparative Fit Index (CFI), and Standardized Root Mean Square (SRMR). CFI values close to .90 and .95 indicate adequate and excellent model fit, respectively, while SRMR values close to .08 and .05 indicate adequate and excellent model fit, respectively [54]. Second, the reliability of the subscales was estimated using McDonald's omega coefficient (ω, [55]).

Next, a series of nested CFA models were estimated to examine measurement invariance by gender and level of sport competition. Specifically, we systematically assessed configural (i.e., whether the factor structure of the AAIS-FR differed by gender or level of sport competition), metric (i.e., whether factor loadings differed across groups), and scalar (i.e., whether intercepts

differed across groups) invariance in a series of increasingly constrained nested models [56]. Model fit of the initial configural model was assessed using CFI and SRMR. Measurement invariance was estimated from changes in the goodness-of-fit indices. Measurement invariance between models is indicated when ΔCFI and ΔSRMR of the later model compared to the previous model are ≤ 0.010 and ≤ 0.015, respectively [57].

After confirming the validity, reliability, and invariance of the AAIS-FR, analyses of variance (ANOVAs) and bivariate correlations were used to examine the concurrent validity using relationships between the two identities, gender, level of sport competition, time spent in academic and athletic contexts, satisfaction with academic and athletic results. Finally, two separate hierarchical multiple regression analyses were conducted to test whether the two identities and their interaction predicted athlete engagement and burnout after controlling for relevant variables correlated with SAs' well-being. Specifically, in a first step, we entered the control variables; then in a second step, we added the two identities to test whether athletic identity was a predictor of sport well-being; and finally, in a third step, we added the interaction term between the two identities to test whether academic identity was a moderator of the relationship between athletic identity and sport well-being.

## Results

### Descriptive statistics

The normality of the data was screened for ABO-S, AEQ and AAIS-FR. The results showed that the majority of skewness values were comprised between -1 and 1 and kurtosis values were comprised between -2 and 2. Only four exceptions with large deviations (greater than 0.5 for skewness or greater than 1.0 for kurtosis) were found: one in the AAIS-FR (item 4 of the athletic identity subscale), one in the ABO-S (item 4 of sport devaluation), and two in the AEQ (items 3 and 4 of enthusiasm). Since most of the items can be considered normal, we used maximum likelihood estimation in the CFAs [58].

### CFA and reliability

The 11-item AAIS-FR two-factor model showed a sub-optimal model fit to the data, $\chi^2$ (43, $N = 359$) = 310, $p < 0.001$, CFI = 0.89, SRMR = 0.06. Examination of standardized factor loadings and omega coefficients indicated that item ATI4 had a low factor loading ($\lambda = 0.38$) and that removal from the scale improved the omega coefficient (from 0.87 to 0.89). A second CFA was performed with 10 items and showed an improvement in the fits, $\chi^2$ (34, $N = 359$) = 259, $p < 0.001$, CFI = 0.91, SRMR = 0.05. Nevertheless, the modification indices (MI) showed the existence of a large residual between a pair of items of the same subscale (i.e., $mi_{ATI5-ATI6}$ = 114.38). Therefore, a third model was run that specified a correlation between the errors of these two items. This model fit the data much better, $\chi^2$ (33, $N = 359$) = 130, $p < 0.001$, CFI = 0.96, SRMR = 0.04. The factor loadings of this model ranged from 0.76 to 0.85 for academic identity and from 0.63 to 0.94 for athletic identity (Table 2). The two subscales were weakly positively correlated ($\phi = 0.29$). The omega coefficients were 0.90 and 0.89 for academic and athletic identity, respectively, indicating adequate reliability.

### Measurement invariance

**Gender invariance.** The configural invariance model fit the data well (Table 3), indicating that the 10-item AAIS-FR factor model was the same for both male and female SAs. When the metric invariance model was compared to the configural invariance model, the changes of CFI and SRMR were within the recommended cut-off, indicating that the metric invariance

**Table 2. Means, standard deviations, skewness, kurtosis and Factor loadings of the final CFA model for each item of the AAIS-FR.**

| Subscale and Item | *M* | *SD* | Skewness | Kurtosis | Factor loadings |
|---|---|---|---|---|---|
| ACI1 –Être un.e étudiant.e compétent·e. *(Being a capable student.)* | 4.16 | 1.16 | -0.31 | -0.24 | 0.76 |
| ACI2 –Être satisfait·e de mon travail universitaire. *(Being satisfied with my academic work.)* | 4.03 | 1.27 | -0.31 | -0.58 | 0.79 |
| ACI3 –Bien réussir à l'université. *(Doing well in school.)* | 4.33 | 1.24 | -0.47 | -0.42 | 0.85 |
| ACI4 –Avoir de bonnes notes. *(Getting good grades.)* | 4.16 | 1.24 | -0.39 | -0.39 | 0.84 |
| ACI5 –Faire parti·e des meilleur·es étudiant·es. *(Having high GPA.)* | 3.10 | 1.51 | 0.20 | -0.87 | 0.76 |
| ATI1 –Être un·e athlète compétent·e. *(Being a capable athlete.)* | 4.99 | 1.07 | -1.13 | 1.34 | 0.94 |
| ATI2 –Être un·e bon·ne athlète. *(Being a good athlete.)* | 5.00 | 1.10 | -1.14 | 0.92 | 0.93 |
| ATI3 –Être athlétique/sportif·ve. *(Being athletic.)* | 5.23 | 1.00 | -1.45 | 2.28 | 0.70 |
| ATI4 –Être fier·ère d'être un·e athlète. *(Being proud to be an athlete.)* | 5.22 | 1.04 | -1.54 | 2.47 | - |
| ATI5 –Être satisfait·e de mes performances sportives. *(Being satisfied with my athletic achievements.)* | 4.80 | 1.19 | -0.80 | -0.18 | 0.63 |
| ATI6 –Bien réussir lors des compétitions sportives. *(Doing well during sport competitions.)* | 4.91 | 1.21 | -1.00 | 0.26 | 0.68 |

*Note*. English original version appeared in brackets. Factor loadings are given for the final version (without ATI4). ACI = Academic Identity; ATI = Athletic Identity.

between genders was met. When the scalar invariance model was compared with the metric invariance model, the ΔCFI and ΔSRMR were within the recommended cut-off, meaning that the scalar invariance between genders was met. These results demonstrated the measurement invariance of the AAIS-FR across gender, suggesting that the scale can be used for both genders.

**Invariance across levels of sport competition.** Configural invariance was demonstrated by on the fit indices, which indicated that the factor patterns were similar across regional, national, and international levels of sport competition (Table 3). A comparison of the other two models showed that the changes in CFI and SRMR were within the recommended cut-off, indicating that both metric and scalar invariances were met. In short, these results supported the measurement invariance of the AAIS-FR by the sport competition level.

**Table 3. Measurement invariance across gender and level of sport competition.**

| Model | $\chi^2$ | Df | CFI | SRMR | ΔCFI | ΔSRMR |
|---|---|---|---|---|---|---|
| Gender invariance models | | | | | | |
| Configural | 184.236 | 66 | 0.950 | 0.045 | - | - |
| Metric | 190.480 | 74 | 0.950 | 0.050 | 0.000 | 0.005 |
| Scalar | 199.985 | 82 | 0.950 | 0.052 | 0.000 | 0.002 |
| Sport competition level invariance models | | | | | | |
| Configural | 237.231 | 99 | 0.941 | 0.064 | - | - |
| Metric | 251.674 | 115 | 0.942 | 0.073 | 0.001 | 0.009 |
| Scalar | 279.125 | 131 | 0.937 | 0.076 | 0.005 | 0.003 |

## Concurrent validity of the AAIS-FR

The means, standard deviations, and correlations of the study variables are displayed in Table 4. Regarding the evidence of concurrent validity, bivariate significant correlations were found between academic identity and satisfaction with academic results ($r = 0.46$, $p < 0.001$) and time spent on academics ($r = 0.17$, $p < 0.010$). In addition, it was not related to the same variables assessed in the athletic context (all $r \leq 0.05$). Athletic identity was significantly correlated with satisfaction with the athletic results ($r = 0.27$, $p < 0.001$) and time spent on sport ($r = 0.27$, $p < 0.001$). It was also negatively correlated with time spent on academics ($r = -0.16$, $p < 0.010$), but not with satisfaction with academic results ($r = -0.02$).

Two two-way ANOVAs were conducted to examine whether differences in SAs' academic and athletic identities could be explained by gender, sport competition level, or the interaction of gender and sport competition level. Regarding academic identity, a significant main effect for gender emerged, $F(1, 353) = 5.22$, $p < 0.050$, with small effect size ($\eta^2_p = 0.015$); females ($M = 4.08$, $SD = 1.07$) reported higher scores than males ($M = 3.83$, $SD = 1.08$). No significant effects were found for level of sport competition, and the two-way interaction (all $F < 1.35$, $p > .260$, $\eta^2_p \leq 0.008$). Regarding athletic identity, a significant main effect for level of sport competition emerged, $F(2, 353) = 15.78$, $p < 0.001$, with a medium effect size ($\eta^2_p = 0.082$). Post-hoc Tukey tests revealed that SAs competing at the regional level reported lower athletic identity ($M = 4.63$, $SD = 1.01$) compared to those competing at the national ($M = 5.09$, $SD = 0.72$) or international ($M = 5.31$, $SD = 0.75$) level (all $t > 4.02$, $p < 0.001$, with medium and large effect sizes, $d = 0.52$ and $0.80$, respectively). However, SAs in these last two groups showed only small differences ($p = 0.084$, $d = 0.27$). No significant effect was found for gender and the two-way interaction (all $F < 1.79$, $p > .18$, $\eta^2_p \leq 0.005$).

## Student-athletes' identities and sport well-being

Examination of the bivariate correlations between the study variables (Table 4) indicated that two demographic variables (i.e., gender and level of sport competition) were related to SA' well-being. Therefore, they were entered into the first stage of hierarchical multiple regression analyses designed to predict athlete engagement and burnout of SAs (Table 5). Adding the two identities significantly increased the explained variance of engagement, $F(4, 346) = 18.60$, $p < 0.001$, $R^2 = 0.18$, $\Delta R^2 = 0.13$, and burnout, $F(4, 346) = 6.62$, $p < 0.001$, $R^2 = 0.07$, $\Delta R^2 = 0.05$. In both analyses, only gender (i.e., males reported higher well-being than females) and athletic identity significantly predicted SA' well-being ($\beta = 0.38$ and $-0.23$ for engagement and burnout, respectively). The addition of the interaction term between the two identities in Step 3 did not improve the explained variance (All $\Delta F < 1.94$, $p > 0.160$, $\Delta R^2 < 0.01$), meaning that academic identity did not have a moderating effect on the relationship between athletic identity and SAs well-being. Additional analyses were conducted using the AEQ and ABO-S subscales. They revealed the same pattern of results as the overall engagement and burnout scores, namely a positive relationship of athletic identity on the four engagement subscales (ranging from 0.20 to 0.37) and a negative relationship of athletic identity on the three burnout subscales (ranging from -0.14 to -0.27).

## Discussion

SAs must be effective in managing both athletic and academic careers. A critical variable in their adaptation, performance, and well-being is the salience of the identity they develop in each of these two contexts [7]. Having reliable and valid instruments to assess the academic and athletic identities of SAs is therefore crucial. Based on its promising psychometric properties, as demonstrated in several international studies [5, 18, 19, 21], the AAIS [6] is the only

**Table 4. Means, standard deviations and correlations among the study variables.**

| | M | SD | 1 | 2 | 3 | 4 | 5 | 6 | 7 | 8 | 9 | 10 | 11 | 12 | 13 | 14 | 15 | 16 | 17 |
|---|---|---|---|---|---|---|---|---|---|---|---|---|---|---|---|---|---|---|---|
| 1. Gender | - | - | - | | | | | | | | | | | | | | | | |
| 2. Level of study | - | - | 0.12* | - | | | | | | | | | | | | | | | |
| 3. Sport competition level | - | - | 0.12* | 0.27*** | - | | | | | | | | | | | | | | |
| 4. Time dedicated to academic | 25.80 | 11.80 | -0.18*** | -0.02 | -0.14** | - | | | | | | | | | | | | | |
| 5. Time dedicated to sport | 11.70 | 6.38 | 0.09 | 0.17** | 0.51*** | -0.21*** | - | | | | | | | | | | | | |
| 6. Satisfaction with academic results | 4.67 | 1.20 | 0.02 | 0.19*** | 0.06 | -0.04 | 0.05 | - | | | | | | | | | | | |
| 7. Satisfaction with athletic results | 4.70 | 1.27 | -0.00 | 0.03 | 0.14* | 0.01 | 0.03 | 0.16** | - | | | | | | | | | | |
| 8. Academic Identity | 3.96 | 1.08 | -0.12* | 0.07 | 0.05 | 0.17** | -0.04 | 0.46*** | 0.05 | - | | | | | | | | | |
| 9. Athletic Identity | 5.02 | 0.86 | 0.10 | 0.07 | 0.29*** | -0.16** | 0.27*** | -0.02 | 0.27*** | 0.23*** | - | | | | | | | | |
| 10. Sport Burnout | 2.42 | 0.63 | -0.13* | 0.01 | -0.10 | 0.16** | -0.00 | -0.13* | -0.42*** | 0.03 | -0.23*** | - | | | | | | | |
| 11. Reduced Accomplishment | 2.44 | 0.71 | -0.10 | -0.07 | -0.13* | 0.11* | -0.08 | -0.11* | -0.51*** | -0.02 | -0.28*** | .81*** | - | | | | | | |
| 12. Sport Devaluation | 2.12 | 0.72 | -0.08 | 0.07 | -0.02 | 0.08 | 0.05 | -0.11* | -0.32*** | 0.04 | -0.16** | .83*** | 0.57*** | - | | | | | |
| 13. Physical Exhaustion | 2.69 | 0.86 | -0.14** | 0.03 | -0.09 | 0.19*** | 0.02 | -0.10 | -0.23*** | 0.06 | -0.14** | .83*** | 0.48*** | 0.51*** | - | | | | |
| 14. Sport Engagement | 4.05 | 0.54 | 0.15** | 0.00 | 0.19*** | -0.18*** | 0.12* | 0.02 | 0.37*** | 0.01 | 0.39*** | -.44*** | -0.47*** | -0.36*** | -0.27*** | - | | | |
| 15. Confidence | 3.78 | 0.72 | 0.18*** | 0.01 | 0.12* | -0.19*** | 0.14** | -0.02 | 0.36*** | -0.02 | 0.36*** | -.40*** | -0.52*** | -0.30*** | -0.20*** | 0.74*** | - | | |
| 16. Dedication | 4.25 | 0.72 | 0.11* | 0.06 | 0.24*** | -0.13* | 0.19*** | 0.01 | 0.25*** | 0.10 | 0.39*** | -.22*** | -0.27*** | -0.16** | -0.12* | 0.79*** | 0.48*** | - | |
| 17. Vigor | 3.79 | 0.69 | 0.16** | -0.05 | 0.12* | -0.20*** | 0.01 | 0.06 | 0.31*** | -0.03 | 0.27*** | -.41*** | -0.37*** | -0.26*** | -0.36*** | 0.81*** | 0.44*** | 0.50*** | - |
| 18. Enthusiasm | 4.39 | 0.65 | 0.02 | -0.02 | 0.10 | -0.04 | 0.01 | 0.00 | 0.22*** | -0.03 | 0.19*** | -.33*** | -0.28*** | -0.41*** | -0.16** | 0.75*** | 0.36*** | 0.45*** | 0.57*** |

Note.

* $p < 0.05$

** $p < 0.01$

*** $p < 0.001$.

**Table 5. Hierarchical multiple regression predicting sport engagement and burnout from academic and athletic identities and their interaction, controlling for gender and sport competition level.**

| Predictor | Sport engagement | | | | | | Sport burnout | | | | | |
|---|---|---|---|---|---|---|---|---|---|---|---|---|
| | Step 1 | | Step 2 | | Step 3 | | Step 1 | | Step 2 | | Step 3 | |
| | β | t | β | t | β | t | β | t | β | t | β | t |
| Intercept | 3.74 | 45.09** | 2.83 | 16.52** | 2.45 | 4.27** | 2.63 | 26.75** | 3.21 | 15.05** | 4.15 | 5.84** |
| Gender | 0.13 | 2.52* | 0.10 | 1.89t | 0.09 | 1.85t | -0.12 | -2.27* | -0.09 | -1.77t | -0.09 | -1.70t |
| Sport compet. level | 0.17 | 3.23** | 0.07 | 1.33 | 0.07 | 1.35 | -0.08 | -1.52 | -0.02 | -0.38 | -0.02 | -0.42 |
| Academic ID | | | -0.07 | -1.42 | -0.07 | 0.46 | | | 0.08 | 1.39 | 0.07 | -1.14 |
| Athletic ID | | | 0.38 | 7.23** | 0.37 | 2.81** | | | -0.23 | -4.18** | -0.22 | -2.56** |
| Acad. ID × Athl. ID | | | | | -0.04 | -0.71 | | | | | 0.07 | 1.39 |
| F | | 9.57** | | 18.60** | | 14.96** | | 4.26* | | 6.62** | | 5.70** |
| ΔF | | | | 26.24** | | 0.50 | | | | 8.80** | | 1.93 |
| $R^2$ | | 0.052 | | 0.177 | | 0.178 | | 0.02 | | 0.071 | | 0.076 |
| $\Delta R^2$ | | | | 0.125 | | 0.001 | | | | 0.047 | | 0.005 |

*Note.* ID = Identity.

* $p < 0.050$

** $p < 0.010$

t $p < 0.100$

instrument recommended in a recent literature review as having the potential to become a gold standard for dual career research [7]. In addition, it is important to validate the scale with SAs from different cultures and countries in order to examine the antecedents and consequences of dual identity in different SAs support structures [7]. Therefore, the aims of this study were both to develop a reliable and valid French version of the AAIS to enable studies on this topic with French-speaking participants, and to examine the additive and interactive relationships of the two identities with sport burnout and engagement among French SAs.

## Preliminary evidence of the AAIS-FR validity and reliability

Overall, the validity and reliability of the AAIS-FR were fairly well supported by four sources of evidence. First, CFAs confirmed the positively correlated two-factor structure of the AAIS-FR as in the original instrument. However, in contrast to this later, our results showed that one item (ATI4, "*Being proud to be an athlete*."] made a relatively small contribution to the athletic identity factor, and deleting it improved the fit indices and reliability. This item is the only one that refers to an athletic pride, a strong emotion related to self-worth that may depend on the performance achieved and the involvement in the athlete role [59]. In other words, some French SAs in our study may have reported a low score on this item not because they had a weak athletic identity, but because they were not performing particularly well at the moment, or were not able to fully engage in their role as an athlete. It's also possible that this adjective has a slightly different meaning in French than in English or other languages into which the AAIS has been translated. Deleting this item does not seem to be an important issue, since it has no equivalent in the academic identity subscale. However, while removing this item ensures the validity and reliability of the French version of the AAIS, it does not facilitate comparisons between samples of SAs from different countries. It is preferable to use the same scale for cross-cultural comparability of results. Consequently, further studies with other samples of French SAs are necessary before a decision is made on the definitive removal of the ATI4 item from the scale.

In addition, the modification indices suggested to add a correlation between the errors of two items on the athletic identity subscale, meaning that these items shared something specificities that the other items on the subscale did not: (ATI5, "*Being satisfied with my athletic achievements.*") and (ATI6, "*Doing well during sport competitions.*"). Indeed, it is possible that the response to item ATI5 is perceived as being closely dependent on the response to item ATI6. It should be noted that the same pattern was observed in the Brazilian version of the AAIS [21]. Further studies are also needed to determine whether this is a culture-specific finding and whether these items need to be modified in the future.

Second, measurement invariance tests were conducted to examine the factorial validity of the AAIS-FR as a function of gender and level of sport competition (i.e., regional, national, and international). Consistent with our hypothesis and previous works [6], our results confirmed the configural, metric, and scalar invariance of the instrument across gender and sport competition level, suggesting that the AAIS-FR can be used to reliably measure the academic and athletic identities of French male and female SAs, regardless of their level of sport competition. Third, as in other studies [6, 19, 21], the two subscales of the AAIS-FR showed acceptable reliability, as indexed by McDonalds' omega coefficients $\geq$ .89, confirming adequate internal consistency.

Fourth, evidence for the concurrent validity of the AAIS-FR was provided by highlighting expected relationships between academic and athletic identities and several correlates identified in previous work. Consistent with our hypothesis and most previous work (e.g., [5, 6, 20]), our results showed that SAs competing at the elite and advanced levels (i.e., national and international) had higher athletic identity than those competing at the lowest level (i.e., regional). While the differences between the top two levels and the lowest level were of medium and large effect size, the athletic identity of international level SAs was only slightly higher than that of national level athletes. Thus, there appears to be a ceiling effect at the national level, while it is likely that the investments and demands associated with the international and Olympic levels are greater than those at the national level. In the same vein, correlational analyses showed that athletic identity was positively correlated with time spent in sport and satisfaction with athletic results, as in previous studies (e.g., [6, 16, 23]). This means that the higher the athletic identity, the more time and energy SAs are likely to invest in the sport context. This greater investment, in turn, maximizes the chances of good results and greater satisfaction.

In line with our hypothesis, we found no gender differences in athletic identity. As we have just seen, only the level of competition (and the associated level of investment) was related to this variable. While older studies have reported a higher athletic identity in males than in females (e.g., [25, 26]), more recent findings have shown no such difference (e.g., [6, 13, 27]). This is likely due to the evolution of gender stereotypes and societal expectations regarding women's place in sport, a field that is no longer perceived as conforming to masculine rather than feminine gender roles [28].

As expected, SAs with a high academic identity spent more time on academics and were more satisfied with their academic results. These findings are consistent with previous work showing that SAs who strongly identify with their role as a student put forth more effort [16], report greater commitment in academics [17], and get better grades [13, 19, 27] than others, which may explain their greater satisfaction with academic results found in our study.

Confirming some of the literature [13, 26, 27], we found that academic identity was higher among female than male SAs. However, this finding is not unique to SAs; several previous works have shown the existence of gender differences in academics, with girls being more engaged and generally performing slightly better than boys [60]. Another explanation that has been proposed concerns the fewer professional career opportunities available to women in the sport context [61]. In order to secure a professional future, it might be in their interest to invest

in the academic context and thus develop a stronger academic identity than men [7, 26]. However, the effect size found in our study was small, and several studies have found no differences (e.g., [6]). Therefore, more research seems to be needed before we can come to a firm conclusion on this issue. Interestingly, we found no differences in academic identity as a function of sport competition level. Previous studies have found inconsistent results. Sometimes SAs at the highest levels of competition had lower academic identity than those at lower levels [6], but sometimes the opposite was true [20], likely reflecting the specificity of the SA support system in each of these studies. The results of our study indicate that competing at the international or Olympic level does not come at the expense of developing an academic identity, which is rather good news.

Finally, a cross-domain relationship was found between athletic identity and time spent studying, and more generally between time dedicated to academics and sport. Specifically, the results showed that the higher the athletic identity (or the more time spent studying), the less time SAs spent on their academics. This finding is consistent with some previous studies that have shown that strong athletic identification can have detrimental effects on the academic domain, whether in terms of effort [16] or grades [5, 17]. It is possible that some SAs with a strong athletic identity experience conflict between the two domains, prioritizing (at least in terms of time) the athletic context at the expense of the academic context. However, the effect size of the correlation is small, and the relationship is significant only for time spent on academics and not for satisfaction with academic performance. In addition, there was no significant relationship between academic identity and time spent on sport or satisfaction with academic results. Ultimately, we found little evidence for the existence of cross-domain relationships; our results mainly confirm the hypothesized relationships between domain identification (academic and/or athletic) and certain correlates within that domain (time spent and satisfaction with results).

## Student-athletes' identities and sport well-being

Given the significant amount of time SAs devote to sport, it is important to know how they feel about it [4], based on positive (e.g., engagement) and negative (e.g., burnout) indicators of well-being [34, 35]. Our study is the first to systematically examine the additive and interactive relationships of both identities with SAs sport well-being. The few previous studies have primarily examined the effects of athletic identity. However, both dominant roles of SAs are likely to affect their well-being and therefore should be systematically measured [5–7]. Taking both identities into account also makes it possible to determine whether one of them is used in a privileged way depending on the context, and how they combine to predict well-being [4].

In line with our hypothesis, the results indicated that athletic identity negatively predicted burnout. The more SAs identified with their athletic role, the fewer burnout syndrome they reported, and vice versa. This finding is consistent with most previous studies (e.g., [38–40]). A strong athletic identity could promote motivation to fully engage in sport [8] and mental toughness [40], making it easier to cope with the physical and psychological discomfort associated with the demands of elite sport [40]. Nevertheless, the weight of the beta coefficient is small, suggesting that athletic identity has limited impact in preventing burnout among SAs. Further research is needed to identify the mechanisms by which athletic identity is related to sport burnout and its moderators.

Academic identity was not significantly related to sport burnout. To our knowledge, only one study has examined the relationship between the two identities and burnout [40]. In contrast to our findings, the study revealed that academic identity was a positive predictor of burnout. The authors concluded that SAs with a strong academic identity are vulnerable to sport

burnout due to the demands of excelling in both academics and athletics. It is possible that the SAs in this study were subjected to greater academic demands than those in our study, but it is also possible that the result of the Martin et al. study is merely a statistical artefact (e.g., a suppressive effect), as the correlation between academic identity and sport burnout was negative rather than positive ($r = -0.25$, $p > 0.050$) and the two identities were highly correlated ($r = 0.71$, $p < 0.010$) [40]. The non-significant relationship we found in our study may mean that for some SAs, having a strong academic identity may be something positive in preventing burnout, and for others, it may be something negative. For example, it is possible that for the former, a strong academic identity allows for the development of resources useful for sport (e.g., concentration, planning, stress management), whereas for the latter, the desire to do well academically may conflict with athletic commitment and thus be a source of burnout. Therefore, future work is needed on the mechanisms that may link academic identity to sport burnout, and to identify for who the relationship is positive or negative. In this regard, measuring perceptions of conflict, enrichment or role separation may be a promising avenue [27, 62].

On the other hand, the moderation analysis we conducted was not significant, meaning that the negative relationship between athletic identity and burnout was identical regardless of the level of academic identity. Given the paucity of previous work, we did not formulate a specific hypothesis. Some authors have suggested that identity foreclosure (i.e., a strong athletic identity combined with a weak academic identity) is a risk factor for sport burnout [63], but the results of studies on this topic are rather inconsistent [15]. Based on a holistic perspective, other authors have suggested that the development of multiple identities (e.g., strong athletic and academic identities) provides numerous benefits, especially in terms of SA well-being (e.g., [61, 64]). While our findings did not confirm either of these perspectives, future studies that wish to test them would do well to systematically use the additive and interactive approach we used in this study.

A pattern of results very similar to burnout was found for engagement, namely a positive relationship between athletic identity and engagement, but no significant effect of academic identity or moderating effect. This finding suggests that the stronger the athletic identity, the higher the engagement reported by SAs. Although we were unable to locate any previous studies that have examined the relationship between SA identities and the specific construct of engagement, this result is nonetheless consistent with our hypothesis and previous studies that have measured constructs closer to certain dimensions of engagement, such as enthusiastic commitment [19], achievement [43] or autonomous [44] motivation. Only in one study [4], a negative relationship was found between athletic identity and a positive dimension of sport well-being (i.e., the sport mental health continuum). This unexpected finding was explained by the disruptions caused by the COVID-19 pandemic, which did not allow SAs to benefit from the support and positive social interactions that typically result from sport activities and are associated with well-being. Further studies are needed to examine the additive and interactive relationships between the two SA identities and various indicators of athletic well-being.

Finally, it should be noted that the weight of the relationship between sport identity and well-being was stronger for engagement than for burnout ($\beta = 0.38$ vs. $\beta = -0.23$, respectively) and explained a greater proportion of the variance (17.70% vs. 7.10%, respectively), confirming the view that burnout and engagement are not two opposite sides of the same coin and the importance of using well-being indicators that capture both adaptive and maladaptive aspects of the experience [34, 35]. Previous studies in self-determination theory have shown that a positive antecedent variable, such as psychological need satisfaction, predicted more positive (e.g., vitality) than negative (e.g., exhaustion) outcomes, while a negative antecedent variable, such as need thwarting, predicted the opposite (e.g., [65]). In light of our findings, it appears that sport identity (a positive variable) is a good predictor of sport well-being, catalysing

confidence, commitment, vitality, and sport enthusiasm, but is a poor predictor of ill-being such as burnout. Similarly, Hill and Curran's [66] meta-analysis of the relationship between perfectionism and burnout found that the positive facet of perfectionism (perfectionistic strivings) was weakly and negatively related to burnout (r = -0.14), while the negative facet of perfectionism (perfectionistic concerns) was moderately and positively related to burnout (r = 0.41). Further studies are needed before firm conclusions can be drawn on this issue.

### Limitations and future directions

This study is not without limitations which deserve to be commented on along with suggestions for future research, to advance the AAIS-FR. First, the different variables measured in this study are exclusively self-reported, which may lead to social desirability and common method bias. Future studies would do well to use multiple methods (e.g., observational, reported by others, physiological data) to assess identities, their antecedents, and consequences to avoid these issues. Second, the cross-sectional design of this study did not allow us to examine the directionality of relationships between variables or the extent to which identities change over time. It is likely that academic and athletic identities fluctuate over the course of an academic year depending on the demands (e.g., exam, competition) and commitments within each role [6, 13]. These identity changes may be associated in turn with different levels of well-being. Longitudinal and/or experimental designs should be favoured in future studies to provide information on the evolution of SA identity salience, its causes and/or consequences. Third, participants are drawn from a convenience sample that is not necessarily representative of the SA population. For example, the first two years of college are overrepresented compared to subsequent years. This did not allow us to reliably examine the relationships between academic year and identities. Further studies using appropriate stratification to obtain a representative sample of the SAs population would help advance knowledge on this topic. Finally, because the development of a valid and reliable questionnaire is a cumulative process, future studies should examine other types of validity, such as criterion validity, concurrent and predictive validities with others variables, or test-retest reliability.

### Conclusion

This study is the first to examine the relationships between athletic and academic identities, various correlates, and well-being among French SAs. This is in line with the European Union's dual career guidelines [67], which call for systematic monitoring of the health and well-being of SAs in each EU country. The results provided evidence of construct (i.e., factor structure) and concurrent validity, as well as reliability (i.e., internal consistency) and invariance across gender and sport competition levels of the AAIS-FR. Thus, researchers can use the AAIS-FR to reliably and validly measure the academic and athletic identities of French-speaking SAs (S1 Appendix). In addition, this study is the first to examine the additive and interactive relationships between the two identities and sport well-being. The results showed a positive relationship between athletic identity and sport well-being, particularly with engagement, suggesting that it may be worthwhile to strengthen this variable to improve SAs wellbeing. They also showed that academic identity does not appear to be an obstacle to sport wellbeing and could therefore be strengthened given its positive relationship with time dedicated to academic and satisfaction with academic results. Nevertheless, future studies are needed to provide further evidence of the validity of the AAIS-FR and to examine other variables, such as inter-role conflict and enrichment, in mediating the relationship between identities and wellbeing.

## Supporting information

**S1 Checklist. Human participants research checklist.**
(DOCX)

**S1 Data.**
(XLSX)

**S1 Appendix. The French version of the AAIS.**
(DOCX)

## Acknowledgments

Access to the facilities of the MSH-Alpes SCREEN platform for conducting the research is gratefully acknowledged. We are grateful to the university officials who put us in touch with student-athletes who could be part of this study, to Lise Philippe for her help in understanding the Finnish article used in our references, and to Ilyes Saoudi and Silvio Maltagliati for their help with the statistical analyses.

## Author Contributions

**Conceptualization:** Solène Lefebvre du Grosriez, Sandrine Isoard-Gautheur, Philippe Sarrazin.

**Data curation:** Solène Lefebvre du Grosriez.

**Formal analysis:** Solène Lefebvre du Grosriez, Sandrine Isoard-Gautheur, Philippe Sarrazin.

**Investigation:** Solène Lefebvre du Grosriez.

**Methodology:** Solène Lefebvre du Grosriez, Sandrine Isoard-Gautheur, Philippe Sarrazin.

**Project administration:** Sandrine Isoard-Gautheur, Philippe Sarrazin.

**Software:** Solène Lefebvre du Grosriez.

**Supervision:** Sandrine Isoard-Gautheur, Philippe Sarrazin.

**Validation:** Sandrine Isoard-Gautheur, Philippe Sarrazin.

**Writing – original draft:** Solène Lefebvre du Grosriez, Sandrine Isoard-Gautheur, Philippe Sarrazin.

**Writing – review & editing:** Solène Lefebvre du Grosriez, Sandrine Isoard-Gautheur, Mariya Yukhymenko-Lescroart, Philippe Sarrazin.

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
