## [Decision Letter · Decision Letter 0]

25 Jan 2024

PONE-D-23-39517The adapted French version of the Academic and Athletic Identity Scale (AAIS-FR): Evidence of validity and reliability and relationships with sport well-beingPLOS ONE

Dear Dr. Sarrazin,

Thank you for submitting your manuscript to PLOS ONE. After careful consideration, we feel that it has merit but does not fully meet PLOS ONE’s publication criteria as it currently stands. Therefore, we invite you to submit a revised version of the manuscript that addresses the points raised during the review process.

We look forward to receiving your revised manuscript.

Kind regards,

Charles Martin-Krumm, Ph.D.

Academic Editor

PLOS ONE

Journal Requirements:

Reviewers' comments:

Reviewer's Responses to Questions

**Comments to the Author**

1. Is the manuscript technically sound, and do the data support the conclusions?

Reviewer #1: Yes

Reviewer #2: Yes

2. Has the statistical analysis been performed appropriately and rigorously? 

Reviewer #1: Yes

Reviewer #2: Yes

3. Have the authors made all data underlying the findings in their manuscript fully available?

Reviewer #1: Yes

Reviewer #2: Yes

4. Is the manuscript presented in an intelligible fashion and written in standard English?

Reviewer #1: Yes

Reviewer #2: Yes

5. Review Comments to the Author

Reviewer #1: The current paper has the purpose to present a French version of the AAIS (AAIS-FR), to test its construct validity, reliability, invariance with gender and sport competition level, as well as its concurrent validity and to examine the relationships between the athletic and academic identities and sport well-being.

The work makes an original and significant contribution to the field of research. The research question and goals are clear and valuable. The manuscript contains a good introduction and is adequately divided into relevant chapters. The article contains coherent and academically rigorous methods and arguments. The claims made are appropriate and the findings are properly substantiated and documented. The evidence given is reliable and justifies the conclusions. The discussion reviews the key findings and points of discussion.

The text is clearly written with academically appropriate language. The ideas flow coherently and the technical terms are well defined.

I only have one minor comment:

In general, I recommend to retain an existing scale in its original form whenever it is possible. Before removing the item ATI4, I suggest to firstly correlate the residuals of ATI5 and ATI6 and see what happens. Probably the model fit will improve above the threshold criteria without necessity to remove ATI4. In my opinion, to have one uniform scale in different languages outweighs the small gain in terms of psychometric properties by removing or changing items. A uniform scale also facilitates the cross-cultural comparability of results.

Since one of the authors of the current study is also the author of the original English scale, I leave it to the authors’ consideration to implement my recommendation or not.

Reviewer #2: Hello and thank you for your excellent article.

there's a little problem with the writing standards. Most of the article is based on Vancouver standards, and some is based on APA standards, so we're going to have to harmonize everything.

: p.3 l.58 To this end, Yukhymenko-Lescroart (2014) developed and 59 validated the Athletic and Academic Identity Scale (AAIS)

p.3 l.72 and group membership (Brewer et al., 1993; Caza et al., 2018)

p.5 l. 102 Yukhymenko-Lescroart (2014) provided evidence

P.7 l. 168 : in the meta-analysis by Lochbaum et al. (2022)

I'm going to suggest a minor revision for your work. As far as I'm concerned, if the elements of bibliographic standards are modified, the article is publishable as is.

Bravo

6. PLOS authors have the option to publish the peer review history of their article (what does this mean?). If published, this will include your full peer review and any attached files.

Reviewer #1: **Yes: **Andreas M. Krafft

Reviewer #2: No

---

## [Author Response · Author response to Decision Letter 0]

27 Jan 2024

Response to reviewers for Manuscript No. PONE-D-23-39517

The adapted French version of the Academic and Athletic Identity Scale (AAIS-FR): Evidence of validity and reliability and relationships with sport well-being

Reviewers' comments:

Reviewer #1: The current paper has the purpose to present a French version of the AAIS (AAIS-FR), to test its construct validity, reliability, invariance with gender and sport competition level, as well as its concurrent validity and to examine the relationships between the athletic and academic identities and sport well-being.

The work makes an original and significant contribution to the field of research. The research question and goals are clear and valuable. The manuscript contains a good introduction and is adequately divided into relevant chapters. The article contains coherent and academically rigorous methods and arguments. The claims made are appropriate and the findings are properly substantiated and documented. The evidence given is reliable and justifies the conclusions. The discussion reviews the key findings and points of discussion.

The text is clearly written with academically appropriate language. The ideas flow coherently and the technical terms are well defined.

We would like to thank the reviewer for his very positive assessment of our work in this manuscript.

1. I only have one minor comment:

In general, I recommend to retain an existing scale in its original form whenever it is possible. Before removing the item ATI4, I suggest to firstly correlate the residuals of ATI5 and ATI6 and see what happens. Probably the model fit will improve above the threshold criteria without necessity to remove ATI4. In my opinion, to have one uniform scale in different languages outweighs the small gain in terms of psychometric properties by removing or changing items. A uniform scale also facilitates the cross-cultural comparability of results.

Since one of the authors of the current study is also the author of the original English scale, I leave it to the authors’ consideration to implement my recommendation or not.. 

We thank the reviewer for this suggestion.

Indeed, the fit indices improve when we simply add a covariance between the errors of items ATI5 and ATI6, without deleting item ATI4 [�2 (42, N = 359) = 181, p < 0.001, CFI = 0.94, SRMR = 0.05]. However, the indices are not as good as those of the model in which item ATI4 was removed [�2 (33, N = 359) = 181, p < 0.001, CFI = 0.96, SRMR = 0.04] and the chi-squared difference is highly significant [��2 (9 = N = 359) = 51, p <.001]. Moreover, model fit is not the only parameter to consider. As recommended by Hair et al. (2019) “standardized loading estimates should be .5 or higher, and ideally, .7 or higher, to indicate convergent validity” (p. 663). Here, item ATI4 had a low factor loading (λ = 0.38); in other words, this item makes a very weak contribution to the Athletic Identity subscale, below the thresholds recommended by Hair et al. Furthermore, the internal consistency (i.e., omega coefficient) of the subscale improves when the ATI4 item is removed (from 0.87 to 0.89). Therefore, we believe that, at least for the sample of student-athletes in our study, the analytic strategy we used is the correct one.

Hair, J., Black, W., Babin, B., & Anderson, R. (2019). Multivariate Data Analysis (8th ed.). Andover: Cengage Learning EMEA.

Nevertheless, we recognize that removing an item from the original scale makes cross-cultural comparability of results less straightforward in the case of international studies using the same instrument. Thus, as this is the first study to use a French version of the questionnaire, we cannot definitively conclude that it is necessary to delete this item for the French population. Further studies are needed.

Therefore, we have added the following sentence to the discussion to take into account the reviewer's recommendation (p. 23):

“However, while removing this item ensures the validity and reliability of the French version of the AAIS, it does not facilitate comparisons between samples of SAs from different countries. It is preferable to use the same scale for cross-cultural comparability of results. Consequently, further studies with other samples of French student-athletes are necessary before a decision is made on the definitive removal of the ATI4 item from the scale.”

Reviewer #2: Hello and thank you for your excellent article. There's a little problem with the writing standards. Most of the article is based on Vancouver standards, and some is based on APA standards, so we're going to have to harmonize everything.

p.3 l.58 To this end, Yukhymenko-Lescroart (2014) developed and 59 validated the Athletic and Academic Identity Scale (AAIS)

p.3 l.72 and group membership (Brewer et al., 1993; Caza et al., 2018)

p.5 l. 102 Yukhymenko-Lescroart (2014) provided evidence

P.7 l. 168 : in the meta-analysis by Lochbaum et al. (2022)

I'm going to suggest a minor revision for your work. As far as I'm concerned, if the elements of bibliographic standards are modified, the article is publishable as is.

Bravo

We thank the reviewer for his/her very positive assessment of the manuscript. We also thank him/her for pointing out some problems with the standards of presentation of the reference in the text. Indeed, there were some references that followed APA standards rather than Vancouver standards. We have carefully checked the references in the manuscript and corrected those that were not up to standard. These are clearly highlighted in the manuscript with the title "Revised Manuscript with Track Changes".

---

## [Editor Report · Decision Letter 1]

1 Feb 2024

The adapted French version of the Academic and Athletic Identity Scale (AAIS-FR): Evidence of validity and reliability and relationships with sport well-being

PONE-D-23-39517R1

Dear Dr. Sarrazin,

We’re pleased to inform you that your manuscript has been judged scientifically suitable for publication and will be formally accepted for publication once it meets all outstanding technical requirements.

Kind regards,

Charles Martin-Krumm, Ph.D.

Academic Editor

PLOS ONE
---

## [Editor Report · Acceptance letter]

26 Apr 2024

PONE-D-23-39517R1 

PLOS ONE

Dear Dr. Sarrazin, 

I'm pleased to inform you that your manuscript has been deemed suitable for publication in PLOS ONE. Congratulations! Your manuscript is now being handed over to our production team.

Kind regards, 

on behalf of

Professor Charles Martin-Krumm 

Academic Editor

PLOS ONE